# Prevalence and Long-Term Prognosis of Post-Intensive Care Syndrome after Sepsis: A Single-Center Prospective Observational Study

**DOI:** 10.3390/jcm11185257

**Published:** 2022-09-06

**Authors:** Shigeaki Inoue, Nobuto Nakanishi, Jun Sugiyama, Naoki Moriyama, Yusuke Miyazaki, Takashi Sugimoto, Yoshihisa Fujinami, Yuko Ono, Joji Kotani

**Affiliations:** 1Department of Disaster and Emergency and Critical Care Medicine, Kobe University Graduate School of Medicine, Kusunoki-Cho 7-5-2, Chuo-Ward, Kobe 650-0017, Japan; 2Department of Emergency Medicine, Kakogawa Chuo Hospital, Honmachi 439, Kakogawa 675-8611, Japan

**Keywords:** post-intensive care syndrome, long-term outcome, Barthel Index, Short-Memory Questionnaire, Hospital Anxiety and Depression Scale, delirium, septic encephalopathy

## Abstract

Post-intensive care syndrome (PICS) comprises physical, mental, and cognitive disorders following a severe illness. The impact of PICS on long-term prognosis has not been fully investigated. This study aimed to: (1) clarify the frequency and clinical characteristics of PICS in sepsis patients and (2) explore the relationship between PICS occurrence and 2-year survival. Patients with sepsis admitted to intensive care unit were enrolled. Data on patient background; clinical information since admission; physical, mental, and cognitive impairments at 3-, 6-, and 12-months post-sepsis onset; 2-year survival; and cause of death were obtained from electronic medical records and telephonic interviews with patients and their families. At 3 months, comparisons of variables were undertaken in the PICS group and the non-PICS group. Among the 77 participants, the in-hospital mortality rate was 11% and the 2-year mortality rate was 52%. The frequencies of PICS at 3, 6, and 12 months were 70%, 60%, and 35%, respectively. The 2-year survival was lower in the PICS group than in the non-PICS group (54% vs. 94%, *p* < 0.01). More than half of the survivors had PICS at 3 and 6 months after sepsis. Among survivors with sepsis, those who developed PICS after 3 months had a lower 2-year survival.

## 1. Introduction

Sepsis is a common disease in an intensive care unit (ICU) [1], a life-threatening organ dysfunction that is caused by a dysregulated host response to infection [2]. Worldwide, there are 31.5 million sepsis cases, 19.4 million severe sepsis cases, and up to 5.3 million deaths annually [3]. The Japanese nationwide data indicated a marked increase in the annual incidence of sepsis hospitalizations, although the annual mortality rates and length of hospital stay in patients with sepsis have significantly decreased, owing to the guidelines of the surviving sepsis campaign and improved treatment [4]. Furthermore, the surviving sepsis campaign guidelines-induced advances in critical care medicine and sepsis care have fostered improved survival among patients with sepsis [5]. Approximately 50% of patients with sepsis recover, one-third die within a year, and one-sixth present with severe, persistent impairments [6]. Long-term mortality is often attributed to the “post-sepsis syndrome (PSS)”—a phenomenon characterized by typical physical, medical, cognitive, and psychological issues after recovery from sepsis [7]. Moreover, the post-intensive care syndrome (PICS) involves physical, cognitive, and mental impairments that occur during ICU stay or after ICU/hospital discharge and thereby impair the long-term prognosis of the affected patients [8,9,10,11].

Recently, several epidemiological studies have reported the incidence and clarified the frequency of PICS. However, the frequency of PICS in patients with sepsis, its risk factors, and the impact of PICS on long-term prognosis after discharge from the hospital has not been fully investigated. Understanding the frequency and risk factors of PICS in patients with sepsis and the impact of PICS on long-term prognosis are crucial for implementing early PICS prevention and treatment. Therefore, this study aimed to (1) clarify the frequency and clinical characteristics of PICS in sepsis patients and (2) explore the relationship between PICS occurrence and the survival rate at 2 years. The primary objective was to ascertain the 2-year survival after sepsis.

## 2. Material and Methods

### 2.1. Study Design and Setting

This single-center prospective observational study was conducted in the ICU of Kobe University Hospital, Advanced Acute Care, and Level 1 Trauma Center in Japan. This mixed ICU comprises 20 beds and receives critically ill surgical, medical, and emergency patients. This study was approved by the Institutional Review Board for Clinical Research of Kobe University Hospital (B210116) and is registered in the UMIN Clinical Trial Registry (registration no. UMIN000048312). Patients with sepsis were prospectively enrolled between November 2019 and May 2020. Written informed consent was obtained from all the patients and/or their relatives.

### 2.2. Study Population and Eligibility Criteria

We enrolled patients who were older than 18 years, were admitted to our ICU, and were diagnosed with sepsis based on the SEPSIS-3 criteria [2]. Patients with the following conditions were excluded from the study: (1) cardiopulmonary arrest (CPA) at the time of admission, (2) not wishing to undergo aggressive treatment at the time of diagnosis at the request of the patient or family, and (3) under 18 years of age.

### 2.3. Procedures

Information on the patient’s background characteristics, clinical information, including vital signs, history, acute physiology and chronic health evaluation Ⅱ score, sequential organ failure assessment (SOFA) score, medication, laboratory examination from days 1 to 7, blood culture results, and antibiotic use were collected from electronic medical records. Delirium in the ICU was evaluated using the Intensive Care Delirium Screening Checklist (ICDSC). Moreover, the investigator telephonically interviewed the patients and their families at 3, 6, and 12 months after the diagnosis of sepsis and asked about PICS, prognosis, and causes of death. The Barthel Index (BI) [12,13], the Short-Memory Questionnaire (SMQ) [14], and the Hospital Anxiety and Depression Scale (HADS) were used to assess physical function, cognitive function, and mental health, respectively. Physical impairment was defined as a BI < 85 [10,12,13], cognitive impairment as a Short-Memory Questionnaire score < 40 [15], and mental impairment as a Hospital Anxiety and Depression Scale ≥ 8 [16]. Data were compared after 3 months between two groups: one with physical, mental, and cognitive impairments (PICS group) and the other without these impairments (non-PICS group).

### 2.4. Outcomes

To explore the long-term outcome of patients with sepsis in this study, we specified the survival of patients with sepsis as the primary outcome. Moreover, we ascertained the prevalence of PICS at 3, 6, and 12 months as well as the risk factors of PICS in patients with sepsis to clarify the number of participants who survived sepsis and which types of them had PICS. We also compared clinically important variables for sepsis and PICS [C-reactive protein (CRP), number of lymphocytes, delirium score, and GCS] from day1–7 between PICS and non-PICS groups, because of the flowing reasons. Lymphopenia is associated with poor outcomes in patients with sepsis [17] and persistent lymphopenia after diagnosis of sepsis predicts mortality [18], CRP is the most widely used and studied biomarkers [19], and CRP may be a simple, early marker and a prognostic factor of 30-day mortality as well as prolonged LOS in survivors for sepsis [20]. Delirium and a lower GCS score are characteristic findings of septic encephalopathy, leading to cognitive impairment of PICS [21]. Finally, we explored the impact of PICS on the 2-year survival after sepsis to elucidate the impact of PICS on long-term outcomes of sepsis.

### 2.5. Statistical Analysis

The demographics and outcomes of the participants are presented as medians (interquartile ranges) for continuous variables and as absolute values and percentages for categorical variables. Continuous variables were compared using the Mann–Whitney *U* test for variance with abnormal distribution, and with the Student’s *t*-test for data with normal distribution, after ascertaining the distribution using the Shapiro–Wilk test. Categorical variables were compared using the chi-square test. Intergroup differences in the variables that were measured at several timepoints were assessed using repeated-measures analysis of variance. To determine the independent risk factors of PICS, multivariable analysis was performed using a binary logistic regression model for PICS occurrence at 3 months after discharge, with the clinically relevant variables. A covariance structure that provided the best fit according to Akaike’s information criterion was used in the analysis. The hazard ratio (HR) with a 95% confidence interval (CI) was used to assess the independent contributions of the significant factors. Survival curves were created using the Kaplan–Meier method and were compared using the log-rank test. A *p*-value of less than 0.05 (two-sided) was considered statistically significant, and all the other data were analyzed using EZR ver. 1.55 [22].

## 3. Results

An outline of this study is shown in Figure 1. During the study period, 86 patients with sepsis were admitted in ICU. Among them, nine were excluded because of preexisting CPA at ICU admission (*n* = 4), lack of informed consent (*n* = 3), and age <18 years (*n* = 2). As the study’s enrollment period commenced prior to January 2020, patients with the coronavirus disease were excluded from this study. Finally, 77 patients with sepsis were enrolled in this study; of these, 64 were discharged alive, 13 died in the hospital, and, 3 months after recovery from sepsis, the remaining 54 surviving patients completed a telephonic interview to answer the questionnaire of the PICS survey; 47 and 43 patients completed the procedure at 6 and 12 months, respectively, after recovering from sepsis.

### 3.1. 2-Year Survival after Sepsis

The characteristics of the 77 participants are summarized in Table 1 [median age, 73 years; 65% men; SOFA score 8.0 on admission]. The 2-year survival after sepsis was 47%; 50% of patients died within 18 months after recovery from sepsis (Figure 2). Among sepsis survivors, the causes of death were related to: sepsis 36%; pneumonia, 20%; meningitis, 4%; soft-tissue infection, 4%; unknown, 8%; and others (64%; cancer, 16%; respiratory failure, 12%; heat failure, 12%; renal failure, 8%; senility, 4%; and unknown, 12%) (Table 2).

### 3.2. Prevalence of PICS after Sepsis

The prevalence rates of PICS at 3, 6, and 12 months were 70%, 60%, and 35%, respectively. In the three domains of PICS, physical impairment was the most frequent characteristic (30–63%) at 3, 6, and 12 months after recovering from sepsis. Furthermore, 24% and 13% of participants had two and three PICS impairments, respectively (Table 3 and Figure 3).

### 3.3. Risk Factors for PICS

Compared with those in the non-PICS group, participants in the PICS group were significantly older, comprised more women, and had a higher rate of mechanical ventilation at 7 days (Table 1). Compared to the non-PICS group, higher scores on the ICDSC but lower Glasgow Coma Scale (GCS) scale on days 1–7, were observed in the PICS group (Figure 4). These results suggest that patients with a diminished level of consciousness or with delirium in acute-phase sepsis are more likely to develop PICS after 3 months.

Immunological and inflammatory assessments showed a trend toward decreased lymphocyte counts in peripheral blood on days 1–7 in the PICS group than in the non-PICS group (Figure 4). The CRP levels were lower in the PICS group than in the non-PICS group on Day 1, but were higher in the PICS group than in the non-PICS group on days 3–7, thereby indicating a prolonged inflammatory response in the PICS group (Figure 4).

Bacteriological assessment demonstrated that, compared with the non-PICS group, the PICS group had a higher detection rate of opportunistic pathogens, including methicillin-resistant *Staphylococcus aureus* (MRSA), extended-spectrum beta-lactamase (ESBL)-producing bacteria, and fungi in blood cultures (Table 4). There was no difference in the first use of antibacterial and antifungal agents between the PICS and non-PICS groups (Data not shown).

To explore the risk factors for PICS, we conducted a logistic analysis with clinically valid variables. There was no significant intergroup difference in age, male sex, and SOFA score; however, the odds ratio of the GCS score on Day 7 was 0.49 (95% CI 0.25–01.96, *p* = 0.04; Table 5).

### 3.4. Patients with PICS at 3 Months after Sepsis Showed Poorer Long-Term Outcome

Finally, to address the impact of PICS at 3 months after sepsis on the long-term outcomes, we compared survival at 2 years after sepsis between the PICS and non-PICS groups. The PICS group showed significantly lower survival rates than the non-PICS group (54% vs. 94%, *p* < 0.01; Figure 5), although there was no intergroup difference in the categorization of each domain (physical, cognitive, or mental impairments).

## 4. Discussion

This study showed that the 2-year survival after sepsis was 47% in 77 patients with sepsis, and the prevalence of PICS at 3, 6, and 12 months was 69%, 60%, and 35%, respectively; 37% of participants had two or more PICS impairments. Next, compared to non-PICS survivors, survivors who had PICS 3 months after sepsis showed a higher detection rate of MRSA- and ESBL-producing bacteria and fungi in blood cultures, increased CRP levels, and a decreased GCS score on Day 7 after sepsis, which was an independent risk factor for PICS. Finally, sepsis survivors who had PICS 3 months after sepsis showed a lower 2-year survival than those without PICS. No previous study has indicated that PICS modulates the long-term prognosis of patients with sepsis, and this is the most important point that has been emphasized in this study.

The 2-year survival rate after sepsis in this study is similar to previous papers [23,24]. Surgical ICU survival, in-hospital survival, and the 2-year mortality rate of patients with sepsis have been reported as 60%, 42%, and 33%, respectively [23]. The 5-year mortality rates for hospital survivors were 56.1% for septic shock, 62.1% for sepsis, and 52.4% for severe infections [24]. The 28-day mortality rate was 29.5%, which increased to 55.4% at 3 years after discharge [25]. Gritte et al. reviewed long-term prognoses from multiple studies and reported that patients with sepsis had 90-day, 1-year, 2-year, and 5-year mortality rates of 9–41%, 12–66%, 8–70%, and 21–100% [26]. The 2-year survival rates in our study match those that were previously reported, indicating that the long-term prognosis after sepsis is very low. Therefore, the oft-used 28-day in-hospital mortality rate may not accurately reflect the prognosis of sepsis.

Among ICU patients, sepsis confers the highest readmission rate of 22–27% [27,28] and 33–36% within 30 and 90 days, respectively [29,30]. The most frequent readmission diagnoses of sepsis survivors within 90 days are sepsis, congestive heart failure, pneumonia, acute renal failure, and respiratory failure [31]. Although sepsis itself was 36% cause of death in sepsis survivors in this study, the remaining 64% had cancer and respiratory and heart failure, which can lead to sepsis or latent sepsis. Therefore, sepsis survivors are more likely to develop infectious diseases, including sepsis and chronic diseases, which confers a higher mortality risk.

In recent years, the emergence of PICS has emphasized that survival and discharge are not the only goals for ICU patients, but perhaps only “just the beginning of the journey” [32]. The impaired quality of life of survivors of severe sepsis and septic shock is similar to that of survivors of critical illness [33]. Huang et al. reported that 48% of patients who had recovered from sepsis experienced recurrent sepsis within the last year, and an increase in sensory, integumentary, digestive, breathing, chest pain, and kidney and musculoskeletal problems [34]. Physical functions, such as daily chores, running errands, spelling, reading, and reduced libido, pose increased difficulty [34]. Survivors of severe sepsis have an increased risk for cognitive and functional impairments, and the prevalence of cognitive impairment increased from 6.1% to 16.7% after sepsis [35]; moreover, physical and cognitive dysfunction are associated with an increased risk of adverse mental health outcomes, including the development of major depression in sepsis and respiratory failure [35,36,37,38]. The prevalence of PICS at 3, 6, and 12 months was 64, 64%, and 56% in ICU patients [39,40]. In COVID-19, prevalence of PICS at 3 months was 75% [41]. Similar trends were observed in this study: the prevalence of PICS at 3, 6, and 12 months was 70%, 60%, and 35%, respectively. In the three domains of PICS, physical impairment was most frequent (30–63%) at 3, 6, and 12 months after sepsis. Huang et al. showed that sepsis survivors experience a significant decrease in physical function, including in their ability to normally perform activities of daily living [34]. These findings suggest the importance of physical rehabilitation. Survivors who received rehabilitation within 3 months of discharge from the hospital had a significantly lower 10-year mortality risk than those who did not, and the frequency of rehabilitation received by survivors was inversely related to mortality [42]. Despite no difference in the frequency or type of rehabilitation between the groups of the survivors and deceased in this study, the PICS group had a significantly higher rate of withdrawal on Day 7 than the non-PICS group (data not shown).

Compared with the non-PICS group, the PICS group had lower GCS scores and higher delirium scores. Furthermore, the GCS score on Day 7 was an independent prognostic factor for PICS onset by 3 months after sepsis. As delirium and a lower GCS score are characteristic findings of septic encephalopathy [21], the PICS group was more likely to have had a higher incidence of septic encephalopathy (SAE), which is a diffuse brain dysfunction that occurs secondary to infection in the body and without an overt CNS infection, with symptoms that range from delirium to deep coma [43]. The international Delirium Epidemiology in Critical Care Study, which included 497 ICU patients showed that the prevalence of delirium was 32.3% and that sepsis was the leading cause of medical illnesses that required ICU admission [44]. More importantly, SAE can cause long-term cognitive dysfunction [45], which persists in 45% of patients at 1 year after discharge. SAEs have been shown to increase the risk of suicide within 2 years of recovery [46], cause long-lasting neurocognitive disturbances (including anxiety or depression), and possibly occur due to inflammatory processes in the brain during sepsis caused by oxidative stress, inflammatory factors, blood–brain barrier injury, changes in cerebral circulation, and emboli of micro vessels [47]. Taken together, the cognitive and psychiatric disturbances that characterize PICS may reflect SAE and drastically impact the quality of life and daily activities of patients with sepsis.

Patients with sepsis often have compromised host immune responses that are characterized by lymphopenia [48,49]; in 25% of sepsis patients who were readmitted with sepsis, the same pathogen was isolated, and a previous gram-negative bacteremia and/or same-site infection predisposed to recurrence of sepsis that was caused by the same pathogen [29]. In addition, nosocomial and multidrug-resistant infections increase the odds of respiratory support and are associated with an increased 12-month mortality risk [50]. Furthermore, our study demonstrated that survivors who had PICS 3 months after sepsis showed a lower number of lymphocytes, and a higher detection rate of MRSA- and ESBL-producing bacteria and fungi in blood cultures. We speculate that immunodeficient patients with lymphocytopenia are more likely to develop opportunistic infections, which leads to PICS and death, as previous review of PICS shows immunosuppression is one of the clinical characteristics in ICU survivors with PICS [51].

The strength of this study is that patients who developed PICS in the third month after sepsis had a significantly poorer prognosis at 2 years than those who did not develop PICS. Therefore, the presence or absence of PICS after hospitalization may be a predictor of the long-term prognosis. A study that evaluated short-term syndromes predictive of long-term prognosis is similar to that of one on ICU delirium as a predictor of prognosis in critically ill patients [52]. Unfortunately, PICS occurrence at 3 months after sepsis was not an independent prognostic factor of the 2-year mortality in the Cox proportional regression model analyses with covariances of age, sex, and the SOFA score on admission in this study (PICS at 3 months; OR 3.2, 95% CI 0.8–11.4, *p* = 0.08) because of the small number of cases. Thus, our study has the potential to shed light on the relationship between PICS and long-term outcomes after sepsis.

This study has several limitations. First, this was a single-center prospective and observational study conducted in a small number of septic patients, leading to the conclusions that cannot be generalized. Although PICS should be evaluated in the entire ICU population, this analysis is limited to post-sepsis PICS because of the diversity of ICU patients and as sepsis has the poorest prognosis. Second, we evaluated PICS using a limited number of questionnaires, including the Barthel Index, SMQ, and HADS. Third, the PICS assessment was conducted by telephonic interviews with the patients and their families, without direct medical examination in outpatient follow-up, which could lead to information bias. Fourth, interventions related to PICS prevention, such as nutritional therapy, rehabilitation, and psychotherapy, have not been adequately evaluated. Despite these limitations, this is the first study to assess the relationship between PICS at 3 months after sepsis and the long-term outcomes.

## 5. Conclusions

The prevalence of PICS at 3, 6, and 12 months after sepsis was 70%, 60%, and 35%, respectively. The GCS score on Day 7 was an independent factor for PICS at 3 months after sepsis. Among survivors with sepsis, those who developed PICS after 3 months had a lower 2-year survival rate.

## Figures and Tables

**Figure 1 jcm-11-05257-f001:**
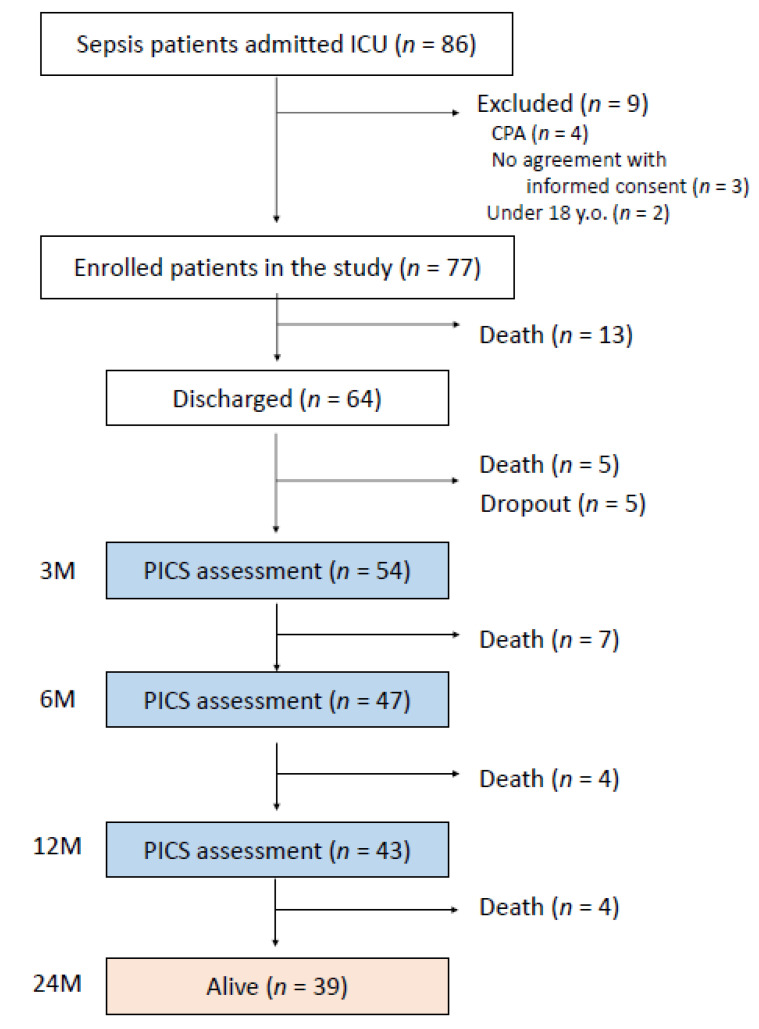
Flowchart of participant enrollment in the study. PICS: post-intensive care syndrome; CPA: cardiopulmonary arrest.

**Figure 2 jcm-11-05257-f002:**
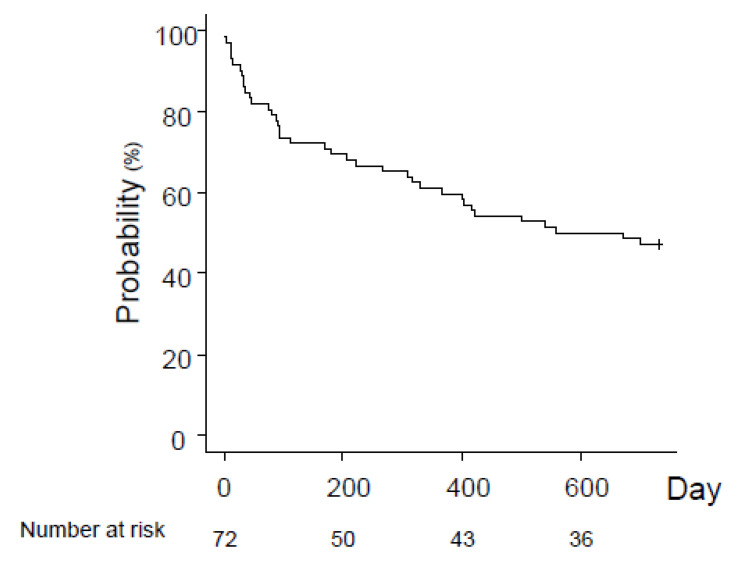
2-year survival after sepsis.

**Figure 3 jcm-11-05257-f003:**
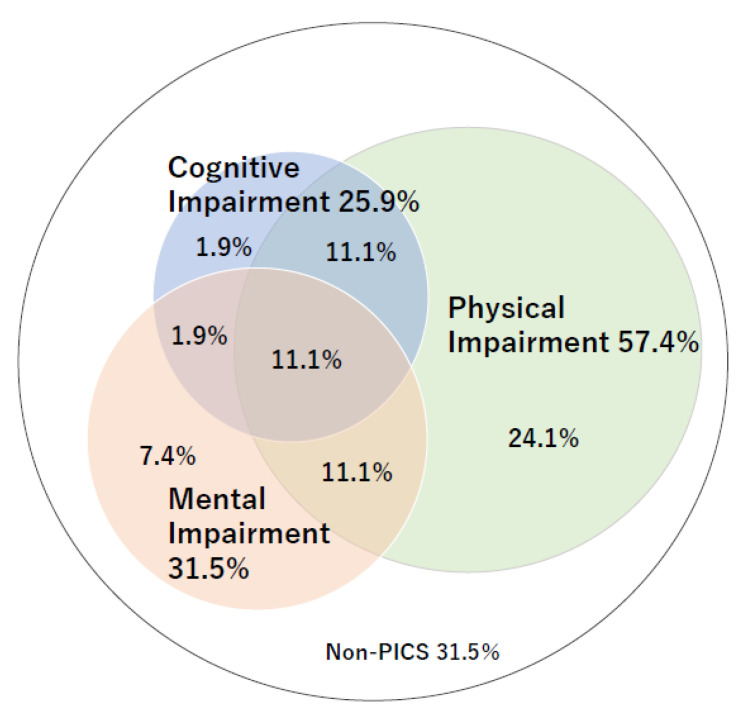
The prevalence of post-intensive care syndrome (PICS) at 3 months after ICU discharge.

**Figure 4 jcm-11-05257-f004:**
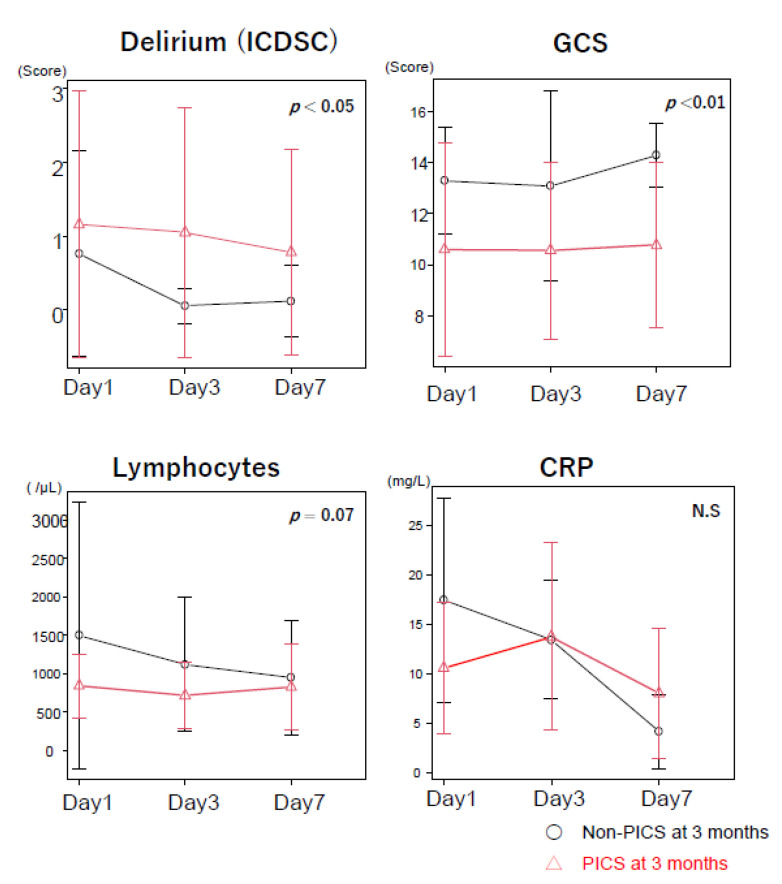
Changes in the ICDSC score, GCS score, number of lymphocytes, and serum CLP levels between participants with PICS at 3 months and those in the non-PICS group at 3 months. Results are presented as mean  ±  SD values. PICS: post-intensive care syndrome; GCS: Glasgow Coma Scale; ICDSC: Intensive Care Delirium Screening Checklist. CRP: C-reactive protein.

**Figure 5 jcm-11-05257-f005:**
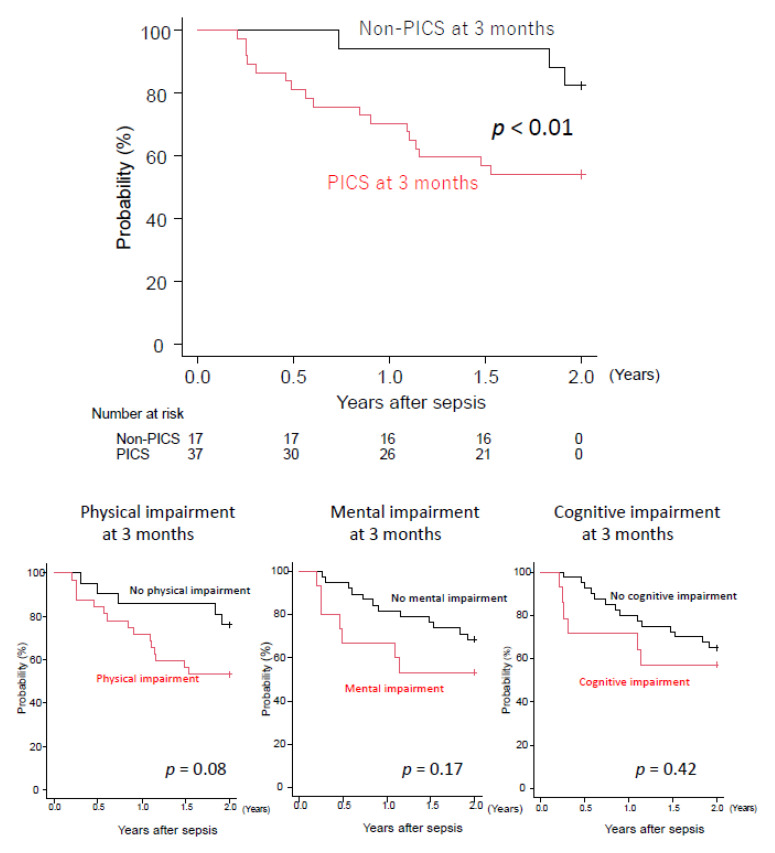
Kaplan–Meier analysis for 2 years of PICS, physical impairment, mental impairment, and cognitive impairment at 3 months after sepsis. PICS: post-intensive care syndrome.

**Table 1 jcm-11-05257-t001:** Characteristics, treatment of sepsis, outcome, PICS assessments of the patients in the study.

		Enrolled Patients	Non-PICS at 3 Months	PICS at 3 Months	*p*. Value
		(*n* = 77)	(*n* = 17)	(*n* = 37)	
Age		73.00 (67.00, 79.00)	70.00 (67.00, 73.00)	77.00 (72.00, 82.00)	0.02
Male (%)		50 (64.9)	15 (88.2)	21 (56.8)	0.03
Race (%)	Japanese	75 (98.7)	17 (100.0)	37 (100.0)	
	Guatemalan	1 (1.3)	0 (0.0)	0 (0.0)	1
Height (cm)		160.70 (147.90, 168.15)	172.00 (162.15, 176.35)	147.80 (144.50, 162.90)	0.11
Body weight (kg)		54.00 (45.05, 60.25)	50.00 (48.25, 58.20)	52.00 (40.90, 54.25)	0.57
**Severity**					
Septic shock on admission (%)		42 (54.5)	6 (35.3)	23 (62.2)	0.08
APACHE II score		20.00 (16.00, 22.00)	19.00 (16.00, 21.50)	18.50 (16.00, 21.00)	0.90
SOFA score at day 1		8.00 (6.75, 11.00)	7.00 (5.25, 9.50)	7.00 (6.25, 9.75)	0.82
SOFA score at day 3		7.00 (5.00, 12.50)	5.50 (4.25, 6.75)	6.00 (5.00, 11.00)	0.61
SOFA score at day 7		6.00 (4.50, 9.75)	6.00 (6.00, 6.00)	4.50 (3.00, 6.00)	0.41
**Comorbidity**					
Hypertension (%)		27 (35.1)	4 (23.5)	16 (43.2)	0.23
Diabetes (%)		22 (28.6)	5 (29.4)	11 (29.7)	1
Cancer (%)		18 (23.4)	5 (29.4)	6 (16.2)	0.29
Autoimmune disorder (%)		15 (19.5)	5 (29.4)	8 (21.6)	0.73
Stroke (%)		15 (19.5)	3 (17.6)	11 (29.7)	0.51
Infection (%)		11 (14.3)	4 (23.5)	5 (13.5)	0.44
Renal failure (%)		9 (11.7)	3 (17.6)	5 (13.5)	0.70
COPD (%)		7 (9.1)	2 (11.8)	3 (8.1)	0.65
Asthma (%)		5 (6.5)	1 (5.9)	2 (5.4)	1
Mental disorder (%)		3 (3.9)	0 (0.0)	2 (5.4)	1
Dementia (%)		3 (3.9)	0 (0.0)	3 (8.1)	0.54
Trauma (%)		2 (2.6)	1 (5.9)	0 (0.0)	0.32
Liver failure (%)		2 (2.6)	0 (0.0)	1 (2.7)	1
Hematological disease (%)		2 (2.6)	2 (11.8)	0 (0.0)	0.10
CNS degenerative disease (%)		2 (2.6)	0 (0.0)	0 (0.0)	1
Endocrine disorder (%)		2 (2.6)	0 (0.0)	1 (2.7)	1
**Medication before admission**					
Immunosuppressant (%)		4 (5.2)	2 (11.8)	0 (0.0)	0.10
Steroids (%)		11 (14.3)	2 (11.8)	6 (16.2)	1
**Causes of sepsis**					
Pneumonia (%)		29 (37.7)	5 (29.4)	16 (43.2)	0.38
Intra-abdominal infection (%)		13 (16.9)	3 (17.6)	7 (18.9)	1
Urinary Tract Infection (%)		10 (13.0)	1 (5.9)	7 (18.9)	0.41
Soft tissue infection (%)		10 (13.0)	3 (17.6)	4 (10.8)	0.67
Burn (%)		3 (3.9)	0 (0.0)	3 (8.1)	0.54
CRBSI (%)		2 (2.6)	0 (0.0)	0 (0.0)	1
**Ventilator**					
at day 1 (%)		35 (45.5)	7 (41.2)	17 (45.9)	1
at day 3 (%)		29 (37.7)	2 (11.8)	15 (40.5)	0.11
at day 7 (%)		22 (28.6)	0 (0.0)	13 (35.1)	0.005
**Emergency operation (%)**		4 (5.2)	2 (11.8)	0 (0.0)	0.07
**Hemoperfusion (%)**		9 (11.7)	2 (11.8)	4 (10.8)	1
**Steroid use (%)**		13 (16.9)	2 (11.8)	6 (16.2)	1
**ICU stay (day)**		4.00 (2.00, 8.00)	3.00 (1.00, 5.50)	4.00 (3.00, 8.00)	0.25
**Death in hospital (%)**		13 (16.9)	0 (0.0)	0 (0.0)	1

APACHE: acute physiology and chronic health evaluation; SOFA: sequential organ failure assessment; CNS: central nervous system; COPD: Chronic Obstructive Pulmonary Disease; CRBSI: catheter related blood stream infection; PICS: Post-intensive care syndrome. Results are presented as number (%) in categorical variables, and median [1st quartile, 3rd quartile].

**Table 2 jcm-11-05257-t002:** PICS assessments of the patients in the study.

	Non-PICS at 3 Months	PICS at 3 Months	*p*. Value
	(*n* = 17)	(*n* = 37)	
**Barthel index (Physical assessment)**			
at 3 months	96.00 (60.00, 100.00)	60.00 (10.00, 92.00)	<0.01
at 6 months	98.00 (40.00, 100.00)	62.50 (10.00, 98.00)	<0.01
at 12 months	98.00 (88.00, 100.00)	70.00 (10.00, 98.00)	<0.01
**HADS (Mental assessment)**			
at 3 months	3.50 (0.00, 6.00)	6.50 (0.00, 14.00)	<0.01
at 6 months	1.50 (0.00, 5.00)	3.00 (0.00, 12.00)	0.08
at 12 months	1.00 (0.00, 2.00)	1.00 (0.00, 12.00)	0.19
**SMQ (Cognitive assessment)**			
at 3 months	46.00 (46.00, 46.00)	46.00 (14.00, 46.00)	<0.01
at 6 months	46.00 (40.00, 46.00)	46.00 (15.00, 46.00)	0.04
at 12 months	46.00 (38.00, 46.00)	46.00 (15.00, 46.00)	0.14

HADS: Hospital Anxiety and Depression Scale; SMQ: Short-Memory Questionnaire; Results are presented as number (%) in categorical variables, and median. [1st quartile, 3rd quartile].

**Table 3 jcm-11-05257-t003:** The prevalence of each PICS domain at 3,6,12 months after ICU admission in septic patients. P: physical impairment; M:mental impairment; C: cognitive impairment.

		P	M	C	Single				Double				Triple	Total
					P	M	C	total	P + M	P + C	C + M	total	P + M + C	
3 months	*n*	31	17	14	13	4	1	18	6	6	1	13	6	37
	%	57.4	31.5	25.9	24.1	7.4	1.9	33.3	11.1	11.1	1.9	24.1	11.1	68.5
6 months	*n*	20	9	10	12	3	4	19	3	3	1	7	2	28
	%	37.0	16.7	18.5	25.5	6.4	8.5	40.4	6.4	6.4	2.1	14.9	4.3	59.6
12 months	*n*	10	5	6	6	2	2	10	1	2	1	4	1	15
	%	18.5	9.3	11.1	14.0	4.7	4.7	23.3	2.3	4.7	2.3	9.3	2.3	34.9

**Table 4 jcm-11-05257-t004:** Detection rate and strains from blood cultures and first antibacterial and antifungal agents used.

	Enrolled Patients	Non-PICS at 3 Months	PICS at 3 Months	*p*. Value
	(*n* = 77)	(*n* = 17)	(*n* = 37)	
**Detection rate of blood culture (%)**				
Day 1	34 (44.2)	7 (41.2)	19 (51.4)	0.57
Day 3–4	6 (7.8)	2 (11.8)	2 (5.4)	0.58
Day 7–8	8 (10.4)	2 (11.8)	5 (13.5)	1
**Strain of bacteria and fungus from blood cultures (%)**				
**GPC**				
*Staphylococcus aureus*	1 (1.3)	0 (0.0)	0 (0.0)	
*Staphylococcus hominis*	1 (1.3)	0 (0.0)	1 (2.7)	
*Streptococcus constellatus*	1 (1.3)	0 (0.0)	1 (2.7)	
*Streptococcus mitis group*	1 (1.3)	0 (0.0)	0 (0.0)	
*Streptococcus pneumoniae*	1 (1.3)	1 (5.9)	0 (0.0)	
*Streptococcus pyogenes*	1 (1.3)	1 (5.9)	0 (0.0)	
*Streptococcus dysgalactiae*	2 (2.6)	1 (5.9)	1 (2.7)	
*Clostridium perfringens*	1 (1.3)	0 (0.0)	1 (2.7)	
Total	9 (11.7)	3 (17.6)	4 (23.5)	0.67
**GNR**				
*Escherichia coli*	6 (7.8)	4 (23.5)	1 (2.7)	
*Klebsiella pneumoniae*	5 (6.5)	1 (5.9)	3 (8.1)	
*Haemophilus influenzae*	1 (1.3)	0 (0.0)	0 (0.0)	
Total	12 (15.6)	5 (6.5)	4 (23.5)	0.19
**Opportunistic pathogen**				
MRSA	5 (6.5)	0 (0.0)	5 (22.7)	
*MRSE*	3 (3.9)	1 (5.9)	1 (2.7)	
*Pseudomonas aeruginosa*	1 (1.3)	0 (0.0)	1 (2.7)	
*Escherichia coli* (ESBL)	4 (5.2)	0 (0.0)	2 (5.4)	
*Enterococcus faecalis*	4 (5.2)	0 (0.0)	3 (8.1)	
*Enterobacter cloacae*	2 (2.6)	0 (0.0)	2 (5.4)	
*Candida albicans*	5 (6.5)	1 (5.9)	2 (5.4)	
*Candida glabrata*	1 (1.3)	0 (0.0)	1 (2.7)	
*Candida tropicali*	2 (2.6)	0 (0.0)	1 (2.7)	
Total	27(35.1)	2 (11.8)	18(48.6)	0.01

Results are presented as number (%); GPC: gram positive coccus; GNR: gram negative rod; MRSA: Methicillin-resistant Staphylococcus aureus; MRSE: Methicillin-Resistant Staphylococcus epidermidis; ESBL: Extended Spectrum beta Lactamase.

**Table 5 jcm-11-05257-t005:** Prediction of PICS at 3 months.

Outcome: PICS at 3 Months			
	Odds Ratio	95% CI	*p*. Value
Age	1.04	0.94–1.14	0.45
Male	1.62	0.094–28.1	0.74
SOFA	1.05	0.75–1.47	0.76
GCS at day 7	0.49	0.25–0.96	0.04

Multiple variable analysis using binary logistic regression model for PICS occurrence at 3 months after discharge; SOFA: sequential organ failure assessment; GCS: Glasgow Coma Scale; CI: confidential interval.

## Data Availability

The datasets used and/or analyzed during the current study are available from the corresponding author on reasonable request.

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
