# Peer review of "Prevalence and Long-Term Prognosis of Post-Intensive Care Syndrome after Sepsis: A Single-Center Prospective Observational Study"

_jcm, 2022, doi:10.3390/jcm11185257_

Round 1

Reviewer 1 Report

This is a study aimed at describing the impact of post-intensive care syndrome (PICS) in a population of septic patients admitted to the ICU.

Even if the topic is current and of interest, I believe some aspects require a better framework:

1) The data presented appear little in line with the literature and raise doubts about the representativeness of the enrolled sample and the reproducibility of the results obtained. In fact, 77 septic patients were enrolled (54.5% with septic shock on admission). Mechanical ventilation is used in 45.5% of cases. The average ICU stay is 4 days and hospital mortality is 16.9%.
Data from the literature suggest that the mortality of patients suffering from septic shock is in the order of 40-50% with a hospital stay certainly greater than that presented.

2) Among the objectives of the work, in addition to the frequency and clinical characteristics of PICS in critically ill patients, the relationship between PICS and survival at 2 years and, overall, survival at 2 years after sepsis are listed. These estimates require an estimate of the number of sample size needed (power) for the data to be sufficiently reliable. However, it does not seem that the study is designed to evaluate the outcome of sepsis but only to describe the evolution of PICS.

3) Patients 'with coronavisus' were excluded. How many ? Identified how? Only the symptomatological or even the simple positives?

4) 77 patients are included but the table shows the data of 73 (according to the text) or 77 according to the table ...

5) Table 1 is illegible. It would be useful to break it down into a part that reports the population studied and a part relating to the PICS

6) Table 2 and Figure 2 are useless

7) How were the parameters presented in Figure 4 chosen? Or rather what is the scientific rationale for which 7th day ventilation is believed to have an impact on the development of PICS? It seems to me that the rationale supporting the hypothesis that the trend of delirium or GCS in the first 7 days can explain the development of PICS at 3-6-12 months is missing ....

8) What is the rationale with which the bacteriological and antibiotic data used are presented?

Author Response

Reviewer 1

This is a study aimed at describing the impact of post-intensive care syndrome (PICS) in a population of septic patients admitted to the ICU.

Even if the topic is current and of interest, I believe some aspects require a better framework:

1) The data presented appear little in line with the literature and raise doubts about the representativeness of the enrolled sample and the reproducibility of the results obtained. In fact, 77 septic patients were enrolled (54.5% with septic shock on admission). Mechanical ventilation is used in 45.5% of cases. The average ICU stay is 4 days and hospital mortality is 16.9%.
Data from the literature suggest that the mortality of patients suffering from septic shock is in the order of 40-50% with a hospital stay certainly greater than that presented.

Reply: Thank you so much for your comments. As you mentioned, the mortality of patients suffering from septic shock used to be in the order of 40-50% about 20 years ago. Now, the mortality of sepsis is improving. In epidemiological studies in the United States, sepsis mortality improved from 40% in 2000 to 27% in 2007 (1). This trend of sepsis mortality in 2014 shows a downward compared to earlier years (2). Sepsis mortality rate in the nationwide epidemiology study 2017 in Japan was 18.3%, a decrease compared to 2010 (3). In-hospital mortality rate in this study was 16.7%, which is comparable to the mortality rate in the nationwide epidemiology study in Japan. Thus, the mortality rate of sepsis is decreasing worldwide, and the patient population in this study is appropriate.

Annual change in the in-hospital mortality (From Reference #1)

Annual change in the in-hospital mortality (From Reference #3)

References

  1. Kumar G, Kumar N, Taneja A, et al. Nationwide trends of severe sepsis in the 21st century (2000–2007). Chest. 2011;140(5):1223–31.
  2. Rhee C, Gohil S, Klompas M. Regulatory mandates for sepsis care–reasons for caution. N Engl J Med. 2014;370(18):1673–6.
  3. Imaeda T, Nakada TA, Takahashi N, et al. Trends in the incidence and outcome of sepsis using data from a Japanese nationwide medical claims database-the Japan Sepsis Alliance (JaSA) study group. Crit Care. 2021;25(1):338.

2) Among the objectives of the work, in addition to the frequency and clinical characteristics of PICS in critically ill patients, the relationship between PICS and survival at 2 years and, overall, survival at 2 years after sepsis are listed. These estimates require an estimate of the number of sample size needed (power) for the data to be sufficiently reliable. However, it does not seem that the study is designed to evaluate the outcome of sepsis but only to describe the evolution of PICS.
Reply: The small sample size in this study is undeniable and is noted in the limitations. However, in this study, the mortality rate of patients who developed PICS 3 months after sepsis was significantly lower than that of the group that did not develop PICS, as shown statistically, albeit in a small sample size. We believe that this result cannot be dismissed based on the small sample size.

3) Patients 'with coronavisus' were excluded. How many ? Identified how? Only the symptomatological or even the simple positives?
Reply: Actually, COVID-19 patients were not transferred to our hospital when Patients with sepsis were enrolled between November 2019 and May 2020 in this study.

4) 77 patients are included but the table shows the data of 73 (according to the text) or 77 according to the table ...
Reply: We have only described 77 patients’ data. We could not find the data of 73 (according to the text), as you mentioned.

5) Table 1 is illegible. It would be useful to break it down into a part that reports the population studied and a part relating to the PICS
Reply: According to your comment, we have divided Table1, into a part that reports the population studied (Table1) and a part relating to the PICS (Table2).

6) Table 2 and Figure 2 are useless
Reply: According to your comment, we have divided Table2.

7) How were the parameters presented in Figure 4 chosen? Or rather what is the scientific rationale for which 7th day ventilation is believed to have an impact on the development of PICS? It seems to me that the rationale supporting the hypothesis that the trend of delirium or GCS in the first 7 days can explain the development of PICS at 3-6-12 months is missing ....

Reply: We chose four clinically important variables for sepsis and PICS in Figure 4 because of the following reasons. Lymphopenia is associated with poor outcomes in patients with sepsis 1, and persistent lymphopenia after diagnosis of sepsis predicts mortality 2.  CRP is by far the most widely used and studied biomarkers 3, and CRP may be a simple, early marker and a prognostic factor of 30-day mortality as well as prolonged LOS in survivors for sepsis 4. Delirium and a lower GCS score are characteristic findings of septic encephalopathy, leading to cognitive impairment of PICS 5.  We also think that prolonged ventilator periods are a barrier to early rehabilitation and cause physical impairment of PICS.

References

  1. Cilloniz C, Peroni HJ, Gabarrús A, et al. Lymphopenia Is Associated With Poor Outcomes of Patients With Community-Acquired Pneumonia and Sepsis. Open forum infectious diseases. 2021;8(6):ofab169.
  2. Drewry AM, Samra N, Skrupky LP, Fuller BM, Compton SM, Hotchkiss RS. Persistent lymphopenia after diagnosis of sepsis predicts mortality. Shock. 2014;42(5):383-391.
  3. Pierrakos C, Velissaris D, Bisdorff M, Marshall JC, Vincent JL. Biomarkers of sepsis: time for a reappraisal. Crit Care. 2020;24(1):287.
  4. Koozi H, Lengquist M, Frigyesi A. C-reactive protein as a prognostic factor in intensive care admissions for sepsis: A Swedish multicenter study. Journal of critical care. 2020;56:73-79.
  5. Huang Y, Chen R, Jiang L, Li S, Xue Y. Basic research and clinical progress of sepsis-associated encephalopathy. Journal of Intensive Medicine. 2021;1(2):90-95.

According to your suggestion, we have added the following sentences in the method section.

“We also compared clinically important variables for sepsis and PICS (CRP, number of lymphocytes, delirium score, and GCS) from day1-7 between PICS and non-PICS groups, because of the flowing reasons. Lymphopenia is associated with poor outcomes in patients with sepsis [17] and persistent lymphopenia after diagnosis of sepsis predicts mortality [18], CRP is the most widely used and studied biomarkers [19], and CRP may be a simple, early marker and a prognostic factor of 30-day mortality as well as prolonged LOS in survivors for sepsis [20].  Delirium and a lower GCS score are characteristic findings of septic encephalopathy, leading to cognitive impairment of PICS [21].” (L96-102)

8) What is the rationale with which the bacteriological and antibiotic data used are presented?

Reply: Since sepsis is a systemic inflammatory reaction triggered by infection and subsequent multiorgan damage, we think that the causative organism of the infection is important information. For this reason, results of bacteria detected from blood cultures are essential in the study of sepsis, and bacteriological data should be provided to the reader in this article. Appropriate early antimicrobial therapy is also important because it greatly influences the prognosis of sepsis, but in the present results, there was no significant difference between the PICS and non-PICS groups in the antibiotic data, so it was removed as you suggested.

Reviewer 2 Report

Introduction

Line 4  "marked increases in annual incidence (?and mortality ?) of sepsis hospitalizations although mortality rates and length of hospital stay in patients with sepsis have significantly decreased"

Figure 8 Glasgow (G)oma Scale

Author Response

Reviewer 2

Introduction

Line 4  "marked increases in annual incidence (?and mortality ?) of sepsis hospitalizations although mortality rates and length of hospital stay in patients with sepsis have significantly decreased"

Reply: According to your comment, we have delated, “and mortality” in Line43

Figure 8 Glasgow (G)oma Scale

Reply: We have revised as, “Glasgow Coma Scale” in Figure 4 and Table5

Reviewer 3 Report

This report deals with sequential studies of patients with COVID-19 infections. The patients were studied at 3, 6 and 12 month intervals as well as at 2 years. COVID-19 patients had 11% mortality rate in hospitals, and a 2-year mortality rate of 52% (n = 72 patients). The 2-year survival rate was lower in the PICS group (54%) versus the non-PICS group 94%. These data require larger groups (>100 per group) in order to confirm the patterns of the data. The current data should be an impetus for future studies.

Author Response

Reviewer 3

  • Figure 1, 2, and 5 is of poor quality;

Reply: According to your comment, we have modified Figure 1,2,5, with enlarging letters.

  • In Table 1, 4, 5 the number of decimal places should be standardized for value "p"

Reply: We have deleted them according to your suggestion.

  • In the introduction, the authors should elaborate on the description of SOFA and APACHE procedures.

Reply: According to your comment, we have modified the sentence in procedure in the method section as below.

“Information on the patient’s background characteristics, clinical information, including vital signs, history, acute physiology and chronic health evaluation â…¡ score, sequential organ failure assessment (SOFA) score, medication, laboratory examination from days 1 to 7, blood culture results, and antibiotic use were collected from electronic medical records.” (L79-80)

  • I recommend that patients admitted to ICU with sepsis be clearly separated from those who developed sepsis during hospitalization;

Reply: According to your comment, we have reanalyzed our data. In 77 patients, 73 patients hospitalized by sepsis, and only 4 patients has developed sepsis during hospitalization (3; post-surgery, 1; post-trauma). Since the number of the patients who developed sepsis during hospitalization was small, we decided not to separate from the all cohort.

5) Out of 44 references, only 14 can be considered the most up-to-date (not older than 3 years). I recommend adding some recent studies:

  1. a) Malik SS, Maqbool M, Rafi A, Kokab N: Prevalence and outcome of infections in intensive care units of a tertiary care hospital in north India. Crit. Care Innov. 2022; 5 (2): 20-28.
  2. b) Vrettou CS, MantziouV, Vassiliou AG, Orfanos SE, Kotanidou A, Dimopoulou I: Post-Intensive Care Syndrome in Survivors from Critical Illness including COVID-19 Patients: A Narrative Review. Life 2022; 12 (1): 107.
  3. c) Krishna G, Kumar S, Sankar R, Raghu K, Sathynarayana V, Siripriya P: Sequential organ failure assessment and modified early warning score system versus quick SOFA score to predict the length of hospital stay in sepsis patients - accuracy scoring study. Crit. Care Innov. 2021; 4 (4): 9-18.
  4. d) Voiriot G, Oualha M, Pierre A, Salmon-Gandonnière C, Gaudet A, Jouan Y, et al. Chronic critical illness and post-intensive care syndrome: from pathophysiology to clinical challenges. Annals of Intensive Care 2022; 12 (1): 1-14.

Reply: According to your comment, we have added some studies you suggested in the manuscript.

Reviewer 4 Report

1) Figure 1, 2, 5 is of poor quality;

2) In Table 1, 4, 5 the number of decimal places should be standardized for value "p"

3) In the introduction, the authors should elaborate on the description of SOFA and APACHE procedures.

4) I recommend that patients admitted to ICU with sepsis be clearly separated from those who developed sepsis during hospitalization;

5) Out of 44 references, only 14 can be considered the most up-to-date (not older than 3 years). I recommend adding some recent studies:

a) Malik SS, Maqbool M, Rafi A, Kokab N: Prevalence and outcome of infections in intensive care units of a tertiary care hospital in north India. Crit. Care Innov. 2022; 5 (2): 20-28.

b) Vrettou CS, MantziouV, Vassiliou AG, Orfanos SE, Kotanidou A, Dimopoulou I: Post-Intensive Care Syndrome in Survivors from Critical Illness including COVID-19 Patients: A Narrative Review. Life 2022; 12 (1): 107.

c) Krishna G, Kumar S, Sankar R, Raghu K, Sathynarayana V, Siripriya P: Sequential organ failure assessment and modified early warning score system versus quick SOFA score to predict the length of hospital stay in sepsis patients - accuracy scoring study. Crit. Care Innov. 2021; 4 (4): 9-18.

d) Voiriot G, Oualha M, Pierre A, Salmon-Gandonnière C, Gaudet A, Jouan Y, et al. Chronic critical illness and post-intensive care syndrome: from pathophysiology to clinical challenges. Annals of Intensive Care 2022; 12 (1): 1-14.

Author Response

(The authors gave the same response as above.)

Round 2

Reviewer 1 Report

Thank for your precise feedback.

I remain skeptical about the opportunity to publish the manuscript even though at this point it is an editor's decision.

Author Response

Thank you very much for your comment.
